# Implications of Microstructure in Helium-Implanted Nanocrystalline Metals

**DOI:** 10.3390/ma15124092

**Published:** 2022-06-09

**Authors:** James E. Nathaniel, Osman El-Atwani, Shu Huang, Jaime Marian, Asher C. Leff, Jon K. Baldwin, Khalid Hattar, Mitra L. Taheri

**Affiliations:** 1Department of Materials Science & Engineering, Drexel University, Philadelphia, PA 19104, USA; jnathaniel@jhu.edu (J.E.N.II); osman@lanl.gov (O.E.-A.); asher.c.leff.ctr@mail.mil (A.C.L.); 2Department of Materials Science & Engineering, Johns Hopkins, Baltimore, MD 21218, USA; 3Materials Science and Technology, Los Alamos National Laboratory, Los Alamos, NM 87545, USA; 4Department of Materials Science & Engineering, University of California Los Angeles, Los Angeles, CA 90095, USA; shuhuang@ucla.edu (S.H.); jmarian@g.ucla.edu (J.M.); 5Army Research Laboratory, Adelphi, MD 20783, USA; 6Center for Integrated Nanotechnologies, Los Alamos National Laboratory, Los Alamos, NM 87545, USA; jkbal@lanl.gov; 7Center for Integrated Nanotechnologies, Sandia National Laboratories, Albuquerque, NM 87185, USA; khattar@sandia.gov

**Keywords:** extreme environments, radiation effects, ion irradiation, helium bubble, nanocrystalline

## Abstract

Helium bubbles are known to form in nuclear reactor structural components when displacement damage occurs in conjunction with helium exposure and/or transmutation. If left unchecked, bubble production can cause swelling, blistering, and embrittlement, all of which substantially degrade materials and—moreover—diminish mechanical properties. On the mission to produce more robust materials, nanocrystalline (NC) metals show great potential and are postulated to exhibit superior radiation resistance due to their high defect and particle sink densities; however, much is still unknown about the mechanisms of defect evolution in these systems under extreme conditions. Here, the performances of NC nickel (Ni) and iron (Fe) are investigated under helium bombardment via transmission electron microscopy (TEM). Bubble density statistics are measured as a function of grain size in specimens implanted under similar conditions. While the overall trends revealed an increase in bubble density up to saturation in both samples, bubble density in Fe was over 300% greater than in Ni. To interrogate the kinetics of helium diffusion and trapping, a rate theory model is developed that substantiates that helium is more readily captured within grains in helium-vacancy complexes in NC Fe, whereas helium is more prone to traversing the grain matrices and migrating to GBs in NC Ni. Our results suggest that (1) grain boundaries can affect bubble swelling in grain matrices significantly and can have a dominant effect over crystal structure, and (2) an NC-Ni-based material can yield superior resistance to irradiation-induced bubble growth compared to an NC-Fe-based material and exhibits high potential for use in extreme environments where swelling due to He bubble formation is of significant concern.

## 1. Introduction

Structural components exposed to extreme environments in nuclear reactors are subject to degradation due to irradiation damage and elevated temperatures [1]. High-energy particles produced during fission and fusion reactions bombard microstructures of materials and can result in the creation of irradiation-induced defects and the absorption of helium [2,3]. Helium migrates readily in metals by the diffusion of interstitial helium atoms, which generally occurs with very low migration energy—less than 0.1 eV [4]; substitutional helium atoms may also diffuse by a conventional vacancy–exchange mechanism enhanced by the atomic movements of displacement cascades. Due to its extremely low solubility, helium has a proclivity to segregate at microstructural features such as voids and interfaces. Helium atoms and vacancies are known to form stable complexes prone to aggregation with more vacancies and helium atoms, resulting in bubble nucleation. Subsequently, the accumulation of helium during operation can alter mechanical properties and decrease material performance even at low concentrations [2,3]. Radiation-induced swelling, embrittlement, and hardening have all been directly linked to helium-induced morphological changes [5].

Nanocrystalline (NC) materials have intricate microstructures composed of grains less than 100 nm in size and consequently have a large volume fraction of grain boundaries (GBs) compared to their coarse-grain analogs. Molecular dynamics simulations and experiments have demonstrated that GBs can absorb point defects and defect clusters, act as sinks for irradiation-induced defects, and contribute to defect reduction by effectively annihilating defects [6]. Due to their high density of internal boundaries, NC materials are postulated to exhibit enhanced defect absorption capabilities and promote an increased irradiation dose threshold necessary to accumulate a density of defects substantial enough to significantly alter macroscopic material properties. GBs are proven to be more efficient at limiting the diffusion of helium atoms due to 2D trapping as compared to 3D trapping in grain matrices [7]. Moreover, the defect sink and helium-trapping properties of GBs can reduce in-grain helium concentration, thereby restricting bubble formation and growth in the matrix [8,9].

On the charge to produce more advanced nuclear materials, one method is to design and develop bulk-nanostructured alloys with enhanced swelling and creep resistance as well as satisfactory corrosion tolerance. Experimental evidence of the correlation of radiation tolerance with grain size is crucial in investigating the performance of candidate nanomaterials for future nuclear applications and for evolving mesoscale predictive models to deconvolute the significance of underlying mechanisms. From coarse-grain studies, it is generally accepted that austenitic stainless steels possess good creep and fatigue resistance at elevated temperatures compared to ferritic/martensitic steels [10]. Adversely, a major disadvantage of austenitic stainless steels is their susceptibility to significant swelling [11,12,13]; however, nanocrystalline materials often have unique physical and chemical properties as compared to their coarse-grained analogs. When investigating fundamental principles, it is expedient to study basic model metals to avoid non-negligible complexities of compounds. Nevertheless, it is imperative to conduct a comparative study accounting for the variety in clustering efficiencies of the crystal structures as well as the mobilities of defects which affect annihilation and agglomeration. Previous studies have demonstrated in both FCC and BCC metals that NC materials have superior radiation tolerance [14,15,16,17,18,19,20,21,22,23]. When exposed to the same irradiation dose and temperature, it was found that the number density and average radius of cavities in NC Cu was smaller than that in single-crystalline and coarse-grained Cu [24]. Similarly, in a study on tungsten, NC grains exhibited a lower in-grain (matrix) bubble density compared to ultrafine grains (between 100 nm and 1 μm) after helium implantation, as well as preferential bubble formation on the GBs [14]. Yu et al. surmised that the reduced bubble density and smaller bubbles in the NC metals should be credited to the depletion of vacancies from the grain interior resulting from the high density of GBs [17]. 

In this work, thin NC Ni (FCC) and NC Fe (BCC) film samples were chosen as model metals to examine (1) how grain boundary density affects bubble formation and bubble damage in the grain matrices, and (2) compare this effect on one FCC and one BCC model material. Our hypothesis is that the increased grain boundary density will lead to low bubble formation and bubble damage in the grain matrices, but the effect of grain boundary density can also be phase-dependent (FCC vs. BCC). in situ transmission electron microscopy (TEM) coupled with helium bombardment was used to implant helium, and automated crystal orientation mapping (ACOM) was used to characterize the post-irradiation microstructure, defect morphology, and grain orientation. A comparison of grain size versus bubble densities in the NC regime revealed that bubble density increased with grain size in both the Ni and Fe; however, the bubble densities were more than 300% greater in Fe compared to Ni. A kinetic rate theory (RT) model was employed to explain differences in helium–void and helium–interface interaction proclivities. The results indicated that in NC systems, helium atoms are more biased towards GBs in Ni compared to Fe, whereas helium atoms and vacancies are more inclined to bond and form bubbles in the grain matrix in Fe compared to Ni. Thus, contrary to standard belief, our results suggest that an NC-Ni-based material can yield a superior radiation tolerance compared to an NC-Fe-based material, exhibiting a high potential for use in extreme environments where swelling due to helium bubble formation is of significant concern.

## 2. Materials and Methods

Thin NC Ni and Fe films (approximately 100 nm-thick) were deposited via physical vapor deposition onto 〈111〉 NaCl substrates. The films were then floated off in a 50/50, water/ethanol solution onto 3 mm Mo mesh TEM grids. The Ni and Fe samples were then annealed in situ at 800 °C and 600 °C, respectively, with a ramp rate of 50 °C/min using a Gatan 628 single-tilt heating stage in a JEOL 2100 LaB_6_ to promote grain growth, while actively monitoring to obtain a stabilized equilibrium nanocrystalline microstructure. Both films maintained a columnar microstructure, while the Ni film was composed of grains ranging from 10 nm to 100 nm in size, and the Fe film was composed of grains ranging in size from 10 nm to 150 nm.

The samples were irradiated in situ using a Colutron implanter attached to a JEOL 2100 LaB_6_ at the in situ Ion Irradiation TEM at Sandia National Laboratories [25]. Each sample was brought to 425 ± 5 °C (within the recovery temperature range for Ni and Fe) using the Gatan heating stage then bombarded with a 10 keV helium ion beam to achieve optimal helium implantation and minimize transmission (<1% of ions) and sputtering yield (0 atoms/ion). The helium beam impinged upon the samples at a nominal incident angle of 60° to a total damage dose of approximately 12.5 displacements per atom (dpa), an ion fluence of 1.5 × 10^22^ m^−2^ for both Fe and Ni. The damage distribution of the ions was simulated using Stopping Range of Ions in Matter (SRIM 2013) Monte Carlo code (see Figure 1); the mean ion range in Ni was 33 nm deep and 35 nm deep in Fe, implying that helium ions were implanted, not transmitted, in the samples. The average vacancy production calculated was 18 vacancies/ion for Ni and 17 vacancies/ion for Fe. An estimation of the primary damage caused by irradiation using the Kinchin–Pease damage model with a threshold displacement energy of 40 eV resulted in a vacancy production of 0.032 vacancies/ion/nm in Ni and 0.029 vacancies/ion/nm in Fe at peak depth (see Figure 1). The final helium concentration at a maximum implantation depth of 100 nm (as shown in Figure 1) is 1.5 × 10^28^ m^−3^ for both Fe and Ni, respectively, and the He/Fe and He/Ni atoms ratios are approximately 0.18 and 0.16 (in the implanted regions), respectively. 

Post-irradiation, bubble characterization was performed using under-focused bright field Fresnel imaging, a proven method which provides an accurate and prompt means of identifying and measuring bubbles, using a JEOL 2100 LaB_6_ TEM. Defect and grain characterization measurements were conducted using the protocol outlined in [26]; ACOM was performed on regions of interest using NanoMegas ASTAR precession electron diffraction system. ImageJ image processing software was used to measure grain and bubble sizes; the polygon selection tool was used to outline and measure grain areas, and bubble diameters were calculated by drawing a circle around the perimeter of randomly chosen bubbles in each grain and averaging their areas. Only bubbles contained within the interiors of the grains were counted. Grains analyzed for quantification were chosen to be in proximity (in the same TEM grid square) to minimize the effects of varying film thickness, heterogeneity in the helium beam, and imaging artifacts.

## 3. Results

After implantation, in both the Ni and Fe films, small cavities were found throughout the grain interior matrices and the GBs; these cavities were presumed to be helium–vacancy bubbles based on previous studies [3]. Figure 2 contains low-magnification, underfocused bright field TEM images of the post-irradiation microstructure analyzed for bubble density in the Ni and Fe specimens captured at room temperature. The grain size and defect morphology in the images are representative of the specimen regions characterized; the Ni sample had an average grain size of ≈40 nm, and the Fe sample had an average grain size of ≈100 nm. From the ACOM scans, a discernable texture was observed in neither the Ni nor the Fe sample; the grains were oriented heterogeneously and predominantly conjoined by random high-angle GBs. Figure 3a depicts the correlation of grain size and the in-grain matrix bubble density (with error bars indicating standard error) for the Ni and Fe samples; the Fe data is adapted from previous work by the authors [27,28]. The range of grain sizes in the Fe sample exceeded but contained the range of grain sizes in the Ni sample, enabling a comparison of similar grain sizes between the two systems. An increasing logarithmic trend in bubble density with grain area is observed in both samples; however, the magnitude of the bubble density saturation at this fluence in Fe is more than 300% greater than in Ni.

In an earlier manuscript [27], El-Atwani et al. described the Fe defect density plot as having three different regimes based on grain size: regime 1 is composed of grains <3000 nm^2^ whose density increases with grain size, regime 2 contains grains between 3000 nm^2^ and 7500 nm^2^ with scattered densities, and regime 3 encompasses grains >7500 nm^2^ with densities approaching a saturation value and eventually decreasing as grain size continues to increase into the ultrafine grain domain. A similar breakdown is found in the Ni plot but with all regimes shifted to smaller grain size ranges: regime 1 contains grains <1000 nm^2^, regime 2 contains grains between 1000 and 4000 nm^2^, and regime 3 with grains >4000 nm^2^. In regime 3 in the Fe plot, the bubble density approaches saturation value near 0.055 bubbles per nm^2^ at the verge of the NC/ultra-fine grain transition (around 100 nm diameter), whereas in the Ni plot, bubble density saturates between 0.015 and 0.020 bubbles per nm^2^ in grains larger than 4000 nm^2^ (around 63 nm diameter). Bubble density saturation suggests a steady state between defect absorption to interfaces such as GBs and helium–vacancy complex formation and coalescence.

Figure 3b depicts bubble diameter as a function of grain size (with error bars indicating standard error) for the same grains quantified in Figure 3a. The average bubble size in the Fe sample was consistent with an average bubble diameter of 3.0 nm with a standard error of 0.24 nm. The Ni sample did not exhibit a trend with respect to grain size, yet the range of bubble diameters varied relatively greatly with a maximum of 9.2 nm, minimum of 1.3 nm, an average of 2.4 nm, and a standard error of 1.53 nm; notably, outlier grains with bubble sizes greater than twice the average only contained 3 or fewer bubbles, and the grains with the largest bubbles had the fewest (i.e., 9.2 nm and 6.5 nm bubble diameters were observed in two grains containing one bubble each).

Crystal structure and grain size have proven to be influential factors in defect cluster formation. Large-scale molecular dynamics simulations of the irradiation of NC Ni and Fe show both similarities and differences in defect production. In both materials, the NC grain size promoted interstitial-free grain interiors but with different vacancy cluster structures forming during displacement cascades [29]. It has been shown both experimentally and from molecular dynamics simulations that dense cascades in FCC metals result in the formation of small (≈2 nm size) stacking fault tetrahedra (SFT) from the unfaulting of Frank loops produced by the collapse of vacancies into platelets on (111) planes [30,31]. Note, however, that the generally accepted primary knock-on atomic energy threshold needed to form SFT is above the helium ion bombardment energy used in the present work. Additionally, the relatively high stacking fault energy of Ni leads to a weaker unfaulting driving force, reinforcing the lack of formation of SFT. Dense displacement cascades in BCC metals, such as in Fe and V [32], have been shown to result in the production of small, isolated vacancy clusters, in agreement with positron annihilation observations of microvoids in irradiated Fe [33].

Under bombardment conditions resulting in the production of Frenkel pairs with primary knock-on atom (PKA) energies below the SFT-formation threshold, faulted Frank loops of a vacancy nature are expected to form and be visible in the form of “black dot” clusters in the TEM in Ni at the temperature and irradiation parameters in the present study [34]; nonetheless, appreciable amounts of these clusters are not observed in the present work. In BCC Fe, the main product of irradiation-induced displacements are small vacancy clusters of irregular shape, sessile 〈100〉 dislocation loops, and glissile self-interstitial clusters that migrate one-dimensionally [35]. When they grow to sufficient size, these clusters turn into perfect 〈111〉 prismatic dislocation loops with high mobility. Ultimately, in both Ni and Fe, interstitials and loops escape the cascade region and migrate to sinks quickly, leaving small vacancy clusters behind. Unless stabilized, vacancy clusters left behind near the cascade core are thermally unstable and dissolve over time, resulting in delayed recombination at defect sinks and the suppression of volume change due to helium bubble growth.

Helium bubbles in crystals occur when vacancy clusters are stabilized internally with helium atoms, leading to the nucleation of bubbles that are thermally stable and trap helium in the bulk. The capacity of bubble growth in an irradiation environment is governed by several factors including helium availability, vacancy production, prominence of defect sinks, and irradiation temperature. The helium implantation profiles shown in Figure 1 show negligible ion beam transmission through the sample, and an area between ≈10 and 30 nm in depth with high overlap between the ion implantation density and the defect production density. This is expected to lead to favorable helium–vacancy bubble formation, although the portion of helium in the crystal is the result of a process where sinks compete to trap helium according to a set of energetics that controls the kinetics of helium transport and its ultimate fate [19,36].

The vacancy migration energies of Fe and Ni (0.67 eV vs. 1.27 eV, respectively) set the temperature for the onset of stage III kinetics (recovery) during isochronal irradiation studies [37]. Consequently, there is consensus that stage III starts at room temperature in Fe compared to ≈100 °C in Ni. Note that SRIM does not account for temperature, and thus, the Frenkel pair production due to PKA and secondary knock-on damage are comparable in Ni and Fe. Helium atoms injected into Fe and Ni microstructures display very low—almost athermal—migration energies (0.06 eV in Fe [37] and between 0.11 and 0.14 eV in Ni [38]) and are expected to migrate continuously unless they are trapped at defects in the grain matrix or GBs, agglomerate with like atoms, or form helium–vacancy clusters which may also be mobile at temperatures above 300 °C whereupon their coalescence can enhance bubble formation as well.

In a variable temperature implantation study, El-Atwani et al. measured bubble density as a function of grain size in two NC Fe samples irradiated to the same damage dose at two different temperatures [28]. NC Fe was shown to have lower bubble density when irradiated at the higher temperature, 425 °C, compared to the sample irradiated at 300 °C. The authors concluded that at higher temperature, helium–vacancy complexes were mobile, resulting in an increase in bubble coalescence, migration, and absorption at nearby grain boundaries, which led to a smaller number of matrix bubble nucleation sites and decreased bubble density. A variable temperature molecular dynamics study of helium diffusion and clustering in Ni confirmed the creation and annihilation of Frenkel pairs and complex formation at 425 °C, with defect production, migration, and cluster growth accelerating at higher temperatures. However, at temperatures at or below ≈325 °C, defect production was hampered resulting in a higher density of smaller defect clusters [36]. However, helium–vacancy complexes are believed to be immobile in the case of the 300 °C implantation in Fe; applying the same reasoning to the current work, one would expect the bubble density to be higher for the NC Ni sample compared to Fe, yet the results are contradictory to this assumption.

To the first order, the differences in bubble density in Fe and Ni can be rationalized in terms of the partition of helium under irradiation in both materials. As indicated above, helium atom migration is nearly athermal (temperature-independent), so one can understand helium accumulation as a process in which atoms are immediately apportioned among the existing defect sinks. The high GB density existing in both materials due to their NC structure provides a strong driving force for helium absorption. However, vacancy clusters in Fe are well predisposed to capture helium atoms to stabilize themselves and give rise to bubble nuclei, while small faulted loops in Ni display a much lower helium capture propensity. This propensity can be quantified to the first order via the helium–vacancy binding energy, which, as shown in Table 1, is considerably larger for Fe than for Ni. Conversely, the helium–GB dissociation energy (i.e., the trapping propensity of helium atoms from GBs) is similar in both materials, although slightly higher in Ni (Table 1). As such, these energetics point to a situation where helium is partitioned more equitably between bubbles and GBs in Fe, whereas it preferentially accumulates at GBs in Ni. The result is a microstructure where matrix bubbles are more prevalent in NC Fe than in Ni. A detailed rate theory (RT) model providing a numerical framework to the above picture is given in Appendix A. The model captures the insertion of primary damage in the form of Frenkel pairs, helium implantation, point defect recombination, and the formation of bubbles, all subjected to strong grain boundary sinks consistent with the present NC microstructures. The model is parameterized using the energies given in Table 1. As Figure 4a below shows, the accumulation of helium in bubbles with irradiation dose in Fe surpasses that of Ni by a factor of two after 200 s of exposure. At the same time, the amount of helium stored in grain boundaries in Ni reaches a steady state, which is about 1.5 times higher than in Fe. This points to a relative partition of helium atoms that is biased towards grain boundaries versus bubbles in Ni, while the opposite is the case for Fe. An interesting consequence of these results is that contrary to the standard behavior displayed by bulk crystals, helium-implanted Ni may offer a higher helium bubble swelling resistance in the grain matrices than Fe when in nanocrystalline forms.

## 4. Discussion

The results of the current study clearly show that the density and size of bubbles formed are quite different in the two materials and are dependent heavily on microstructure, i.e., the presence of GBs acting as effective sinks. Evidence of crystal structure influencing irradiation damage defect production is apparent in the defect yield in BCC metals being reported as much lower than in FCC metals [44,45,46,47,48]; however, in these previous studies, GB densities were not as prevalent and influential as in the present study. As Singh and Foreman showed [49], GBs and grain size can tilt the balance and diminish the importance of the underlying crystal structure on helium bubble swelling. Indeed, the simulation results of Figure 4 prove that by adding an ultrahigh density of GB sinks, an FCC metal like Ni can be turned into a more helium swelling resistant material compared to Fe considering grain matrices only. Furthermore, contrary to bulk or coarse-grained materials, at smaller length scales the influence of GB-helium trapping can prove to be more dominant than diffusivity.

The microstructural influence of GB density may have more implications in assorted metals and compounds as well as alloying elements—such as Ni, Si, P, Ti, and C—that have varying effects on defect evolution behavior in different alloys [50]. Since helium bubble swelling is a result of the partitioning of irradiation-induced point defects among different sinks, alloy elements in solutes or precipitates will certainly contribute differently to microstructural evolution, which will undoubtedly be affected by grain size and GB character as well. Bubble-denuded zones (BDZ), similar to void-denuded zones [51,52], indicate defect absorption and helium-depleted zones adjacent to interfaces; BDZs have been identified in helium-implanted ferritic metals [28,53]. Complex factors, including intrinsic (matrix texture, GB misorientations [51,54], GB energies [51], etc.) and extrinsic influences (temperature [55], helium concentration [56], etc.), are reportedly known to exert non-negligible effects on BDZ formation and evolution.

In the same vein, the capacity of GBs to limit bubble growth in acute extreme conditions is questionable and requires more thorough investigation. In a sequential self-ion irradiation, helium implantation, and annealing study in NC Ni, Muntifering et al. [57] observed large bubbles with diameters on the order of the grain size and specimen thickness with many cavities spanning multiple grains. The authors concluded that the order of the irradiation and implantation had a significant impact on the resulting damage/bubble distribution and structure and that GBs had a negligible ability to limit the growth of cavities with temperature. Additionally, applied stress is critical to swelling behavior [58], which is also related to irradiation creep, another phenomenon that also involves the flow of irradiation-induced point defects under stress. GB-helium sink behavior invariably leads to bubble nucleation within the GB, which can lead to intergranular embrittlement [59]; furthermore, the density of bubbles in the GBs and GB embrittlement are necessary considerations for future work.

Consideration of the gas atom pressure inside bubbles is known to have relevant implications for material performance; how complex composition affects bubble diffusion and agglomeration are of particular concern. Nevertheless, due to experimental restraints, the stoichiometric ratios of the bubbles resulting from the implantations in the current work were unspecified. Model investigations of helium–vacancy complex (He_m_V_n_) evolution have found that the stable configuration of complex clusters is impartial to cluster size and discrete numbers of helium atoms but dependent on the helium atom to vacancy ratio, m:n [42]. Results from a large-scale accelerated molecular dynamics study emphasize the disparities of diffusivities of He_m_V_n_ clusters as a function of m, the number of helium atoms [60]; the simulation indicated that complexes can interconvert (varying values of m and n) due to Frenkel pair and annihilation events and defect diffusion. At their peak mobility, the diffusivity of these clusters may be comparable to that of a vacancy or up to five orders of magnitude slower depending on the m:n ratio. Molecular dynamics studies on Ni and Fe have also shown that as the number of helium atoms (m) increases in the ratio, the helium binding energy decreases and the complex formation energy increases [37,40,42]. The energetics of helium trapping and bubble growth within GBs with respect to local atomistic structure and environment have also been linked to cluster stoichiometry [43,61]. Additionally, there is evidence of complex chemistry dependence on irradiation parameters [62]. As affirmed by the aforementioned studies, the externalities of helium–vacancy cluster stoichiometry imperatively influence defect morphologies and is worth further experimental ascertainment.

Lastly, extrapolating the implications of the current work, a prudent next step would be the development of a more comprehensive model which delineates the details of helium diffusion and trapping coupled with in situ experimental validation to resolve how elemental components and irradiation parameters affect defect morphologies. Ultimately, modeling and careful experimentation/validation can be used in conjunction as a design tool to calculate the optimum grain size to achieve the maximal radiation tolerance of a particular system.

## 5. Conclusions

In conclusion, the defect morphology resulting from low-energy helium implantation was studied in thin free-standing NC (FCC) Ni and (BCC) Fe films. Bubble density and size were measured and compared with respect to grain size. A decreasing trend in bubble density as grain size decreases is distinct for both samples. Contrary to findings in coarse-grain studies, the bubble density in the grain matrices in Fe was 300% greater than in Ni, suggesting that the NC Ni may possess an enhanced resistance to changes in volume due to helium bubble growth. A kinetic RT model was employed to differentiate between helium–defect and helium–interface interaction proclivities; the model results indicate in that in NC systems (grain size < 100 nm), helium is partitioned more equitably between bubbles and GBs in Fe, whereas it preferentially accumulates at GBs in Ni, resulting in a microstructure where bubbles are more prevalent in NC Fe than in Ni. Moreover, many factors such as complex composition, irradiation parameters, material chemistry, and GB sink efficiency all affect the kinetics and play a significant role in defect evolution and morphology in irradiated NC materials and require further exploration.

## Figures and Tables

**Figure 1 materials-15-04092-f001:**
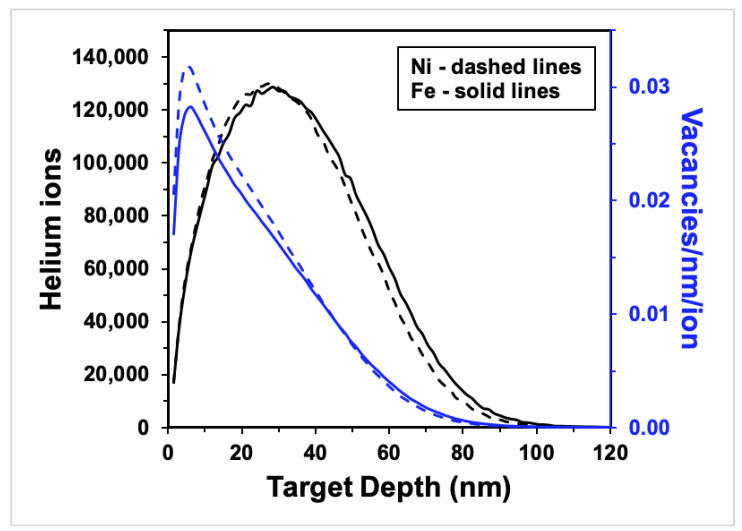
SRIM calculations for 10 keV helium ions bombarding 100 nm-thick Ni (dashed lines) and Fe (solid lines) samples at 60° incidence to the ion beam: ion distribution (black curves) and vacancy production (blue curves) vs. target depth. (For interpretation of the references to color in this figure, the reader is referred to the web version of this article).

**Figure 2 materials-15-04092-f002:**
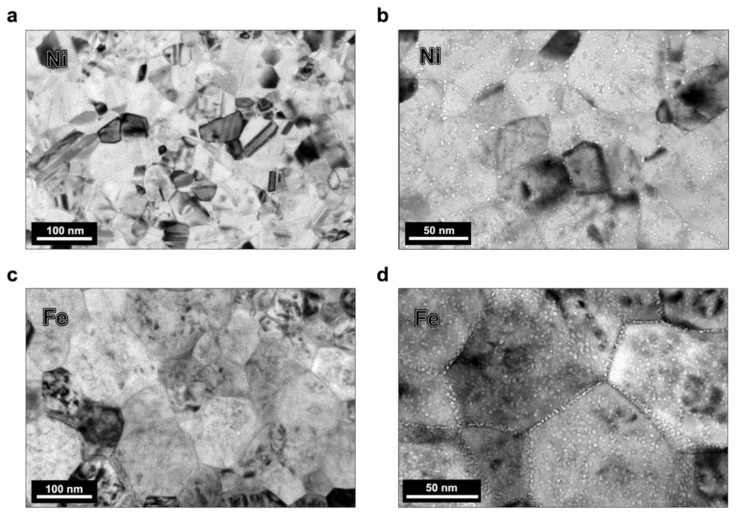
Under-focused bright field TEM images of regions in helium implanted (**a**,**b**) Ni and (**c**,**d**) Fe samples; bubbles are seen with white contrast decorating the grain interiors and GBs.

**Figure 3 materials-15-04092-f003:**
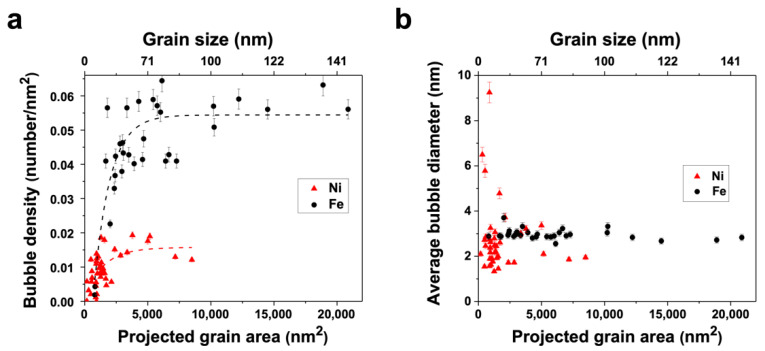
(**a**) Areal bubble density (number/nm^2^) in the grain interiors correlated with grain size (nm^2^) for 10 keV helium-implanted NC Ni (red triangles) and NC Fe (black circles); dotted line fitting demonstrates the trend in density change. (**b**) average bubble diameters correlated with grain area for the NC Ni and NC Fe grains, represented in Figure 3a. Fe data was adapted with permission from [27,28]. Copyright 2017 Elsevier, Copyright 2017 Taylor & Francis.

**Figure 4 materials-15-04092-f004:**
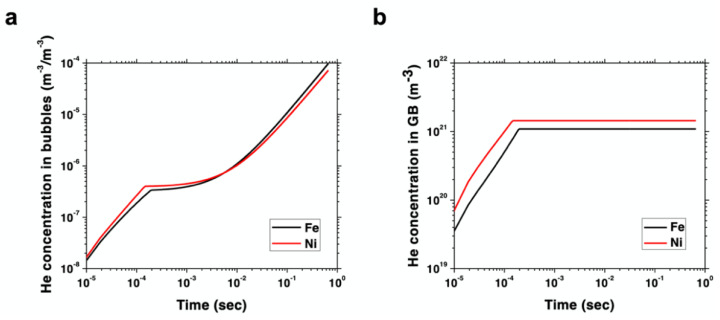
Rate theory model results depicting (**a**) concentration of helium (in m^–3^) per unit bubble volume (also in m^–3^) as a function of time during irradiation in NC Ni and Fe under the conditions of this study and (**b**) accumulation of helium atoms (in m^–3^) at grain boundaries in Ni and Fe as a function of time.

**Table 1 materials-15-04092-t001:** Table of binding and migration energies.

	*Vacancy Migration Energy (eV)*	*Helium Migration Energy (eV)*	*Helium–Vacancy Binding Energy (eV)*	*Helium–GB Binding Energy (eV)*
**Ni**	1.27 eV [39]	0.11–0.14 eV [38]	3.5 eV [40]	1.0 eV [41]
**Fe**	0.67 eV [37]	0.06 eV [37]	5.0 eV [42]	0.8 eV [43]

## Data Availability

The data presented in this study are available on request from the corresponding author.

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
