# Peer review of "Implications of Microstructure in Helium-Implanted Nanocrystalline Metals"

_materials, 2022, doi:10.3390/ma15124092_

Round 1
Reviewer 1 Report
The manuscript by M.L. Taheri et al. investigates the resistance of nanocrystalline Ni and Fe with respect to He bubble formation under ion bombardment, using TEM and a kinetic model. They find that the bubble density is higher in Fe with respect to Ni, which shows higher resistance to bubble growth. This is attributed to a different grain boundaries density in the two materials.
The paper shows good quality data and an exhaustive discussion, based also on the kinetic model developed by the authors (and shown as supplementary material) to explain the different inclination of He to form bubbles in iron with respect to nickel.
The paper may be useful as a contribution in the discussion about innovative materials to be considered in nuclear reactors; it can be published in its present form. I only suggest the authors to explain the acronym dpa (line 124) - unclear to a reader not familiar with radiation damage - if we consider that all the other acronyms were clarified.
The paper by M. L. Taheri et al. aims to investigate the resistance of two nanocrystalline materials (Ni and Fe) to He bubble formation, a well-known problem in nuclear plants affecting the structural components. The main idea is that the higher density of grain boundaries in the investigated materials with respect to coarse grain analogues could reduce the bubble damage. This idea is not original, but according to my knowledge it is the first time that it is applied to compare the performances of Ni and Fe.
The experimental technique adopted (TEM) is a robust method to identify the presence of bubbles. The number of figures and tables is reduced to a minimum, but sufficient to support the long discussion of the results. This is in my opinion preferable with respect to an excess of data, which sometimes appear both in tables and in figures. The discussion is clear and complete, supported by a kinetic model developed and shown in a supplementary material. The conclusions point out a better resistance of Ni with respect to Fe, although other factors, not taken into consideration in the present work, can influence the defect. The references are complete and up-to-date and the number of self-citations of the authors (six over 64) is reasonable.
The topic is not original, it has been tackled by several authors (as shown in the references), but the data displayed are worth to be published. For this reason, I considered the quality of the paper (originality, significance, soundness) as 'average' in my previous report. Nevertheless, it is well written, with precisely defined goals, with good quality data and a sound discussion.
For these reasons I proposed to publish the paper with a minor correction.
Author Response
We genuinely appreciate the reviewer's assessment and feedback on the manuscript. We have addressed the reviewer's recommendation to explain the acronym (dpa) in an edited version of the manuscript which we will upload and resubmit. Thank you for your time and attention.
Reviewer 2 Report
The authors studied the effect of helium implantation on the bubble formation of nanocrystalline materials and investigated an NC Fe as a model material for bcc metal, and an NC Ni as a model for fcc metal. The results are interesting, and I recommend the publication of this manuscript after addressing the following concerns:
1. The TEM images shown in Fig. 2 present the structure of Ni and Fe after He implantation, but the grain size of these two samples is not the same, which should be considered.
2. The bubble density/size in Ni/Fe are compared as a function of grain size, TEM evidence should be provided. It will be clearer to the reader if the in-situ TEM testing images are presented.
3. Grammar should be carefully checked throughout the manuscript.
For example, on Page 2 Line 93, the “1)” should be “2)”; on Page 10 “Helium-vacancy cluster, i.e. bubble, density and size analysis were conducted with respect to grain size.” This sentence is difficult to read.
Author Response
We appreciate the reviewer's attention to and feedback on the manuscript. The reviewer's recommendations have been addressed in an edited version of the manuscript which will be uploaded and resubmitted.
To address the reviewer's concern about the range of grain sizes in the specimens not being represented, Fig. 2 has been edited to include two more images representative of the Ni and Fe specimens' final microstructure and defect morphology.
Pertaining to the reviewer's comment about the lack of in situ micrographs: The in situ experiment was conducted with the specimens in focus to monitor for changes in grain structure (grain growth, GB migration), dislocation movement, and potential oxide growth. However, bubbles are only visible when viewing "through focus," i.e. when the specimen is either over-focus or under-focus, hence bubble analysis was conducted post implantation. We agree with the reviewer, it is worthwhile to compare bubble evolution throughout implantation and would be an interesting experiment for a future study.
Lastly, the authors have taken the reviewer's advice and have proofread the manuscript and made corrections to grammatical errors including the two examples identified in the reviewer's response.
Again, we would like to thank the reviewer for their time and helpful suggestions.
Round 2
Reviewer 2 Report
I recommend publication of the updated manuscript.